# TESTAM: A Time-Enhanced Spatio-Temporal Attention Model with Mixture of Experts

**Hyunwook Lee & Sungahn Ko**[*]
Ulsan National Institute of Science and Technology
{gusdnr0916, sako}@unist.ac.kr

## Abstract

Accurate traffic forecasting is challenging due to the complex interdependencies of large road networks and abrupt speed changes caused by unexpected events. Recent work has focused on spatial modeling with adaptive graph embedding or graph attention but has paid less attention to the temporal characteristics and effectiveness of in-situ modeling. In this paper, we propose the time-enhanced spatio-temporal attention model (TESTAM) to better capture recurring and non-recurring traffic patterns with mixture-of-experts model with three experts for temporal modeling, spatio-temporal modeling with a static graph, and spatio-temporal dependency modeling with a dynamic graph. By introducing different experts and properly routing them, TESTAM better captures traffic patterns under various circumstances, including cases of spatially isolated roads, highly interconnected roads, and recurring and non-recurring events. For proper routing, we reformulate a gating problem as a classification task with pseudo labels. Experimental results on three public traffic network datasets, METR-LA, PEMS-BAY, and EXPY-TKY, demonstrate that TESTAM outperforms 13 existing methods in terms of accuracy due to its better modeling of recurring and non-recurring traffic patterns. You can find the official code from https://github.com/HyunWookL/TESTAM

## 1 Introduction

Spatio-temporal modeling in non-Euclidean space has received considerable attention since it can be widely applied to many real-world problems, such as social networks and human pose estimation. Traffic forecasting is a representative real-world problem, which is particularly challenging due to the difficulty of identifying innate spatio-temporal dependencies between roads. Moreover, such dependencies are often influenced by numerous factors, such as weather, accidents, and holidays (Park et al., 2020; Lee et al., 2020; Lee et al., 2022).

To overcome the challenges related to spatio-temporal modeling, many deep learning models have been proposed, including graph convolutional networks (GCNs), recurrent neural networks (RNNs), and Transformer. Li et al. (2018) have introduced DCRNN, which injects graph convolution into recurrent units, while Yu et al. (2018) have combined graph convolution and convolutional neural networks (CNNs) to model spatial and temporal features, outperforming traditional methods, such as ARIMA. Although effective, GCN-based methods require prior knowledge of the topological characteristics of spatial dependencies. In addition, as the pre-defined graph relies heavily on the Euclidean distance and empirical laws (Tobler's first law of geography), ignoring dynamic changes in traffic (e.g., rush hour and accidents), it is hardly an optimal solution (Jiang et al., 2023). Graph-WaveNet, proposed by Wu et al. (2019), is the first model to address this limitation by using node embedding, building learnable adjacency matrix for spatial modeling. Motivated by the success of Graph-WaveNet and DCRNN, a line of research has focused on learnable graph structures, such as AGCRN (Bai et al., 2020) and MTGNN (Wu et al., 2020).

Although spatial modeling with *learnable static graphs* has drastically improved traffic forecasting, researchers have found that it can be further improved by learning networks dynamics among time, named *time-varying* graph structure. SLCNN (Zhang et al., 2020) and StemGNN (Cao et al.,

---

[*]Corresponding author

2020) attempt to learn time-varying graph structures by projecting observational data. Zheng et al. (2020) have adopted multi-head attention for improved dynamic spatial modeling with no spatial restrictions, while Park et al. (2020) have developed ST-GRAT, a modified Transformer for traffic forecasting that utilizes graph attention networks (GAT). However, time-varying graph modeling is noise sensitive. Attention-based models can be relatively less noise sensitive, but a recent study reports that they often fail to generate an informative attention map by spreading attention weights over all roads (Jin et al., 2023). MegaCRN (Jiang et al., 2023) utilizes memory networks for graph learning, reducing sensitivity and injecting temporal information, simultaneously. Although effective, aforementioned methods focus on spatial modeling using *specific spatial modeling methods*, paying less attention to the use of multiple spatial modeling methods for in-situ forecasting.

Different spatial modeling methods have certain advantages for different circumstances. For instance, learnable static graph modeling outperforms dynamic graphs in recurring traffic situations (Wu et al., 2020; Jiang et al., 2023). On the other hand, dynamic spatial modeling is advantageous for non-recurring traffic, such as incidents or abrupt speed changes (Park et al., 2020; Zheng et al., 2020). Park et al. (2020) have revealed that preserving the the road information itself improves forecasting performance, implying the need of temporal-only modeling. Jin et al. (2023) have shown that a static graph built on temporal similarity could lead to performance improvements when combined with a dynamic graph modeling method. Although many studies have discussed the importance of effective spatial modeling for traffic forecasting, few studies have focused on the dynamic use of spatial modeling methods in traffic forecasting (i.e., in-situ traffic forecasting).

In this paper, we propose a time-enhanced spatio-temporal attention model (TESTAM), a novel Mixture-of-Experts (MoE) model that enables in-situ traffic forecasting. TESTAM consists of three experts, each of them has different spatial modeling: 1) without spatial modeling, 2) with learnable static graph, 3) with with dynamic graph modeling, and one gating network. Each expert consists of transformer-based blocks with their own spatial modeling methods. Gating networks take each expert's last hidden state and input traffic conditions, generating candidate routes for in-situ traffic forecasting. To achieve effective training of gating network, we solve the routing problem as a classification problem with two loss functions that are designed to avoid the worst route and lead to the best route. The contributions of this work can be summarized as follows:

- We propose a novel Mixture-of-Experts model called TESTAM for traffic forecasting with diverse graph architectures for improving accuracy in different traffic conditions, including recurring and non-recurring situations.

- We reformulate the gating problem as a classification problem to have the model better contextualize traffic situations and choose spatial modeling methods (i.e., experts) during training.

- The experimental results over the state-of-the-art models using three real-world datasets indicate that TESTAM outperforms existing methods quantitatively and qualitatively.

## 2 RELATED WORK

### 2.1 TRAFFIC FORECASTING

Deep learning models achieve huge success by effectively capturing spatio-temporal features in traffic forecasting tasks. Previous studies have shown that RNN-based models outperform conventional temporal modeling approaches, such as ARIMA and support vector regression (Vlahogianni et al., 2014; Li & Shahabi, 2018). More recently, substantial research has demonstrated that attention-based models (Zheng et al., 2020; Park et al., 2020) and CNNs (Yu et al., 2018; Wu et al., 2019; 2020) perform better than RNN-based model in long-term prediction tasks. For spatial modeling, Zhang et al. (2016) have proposed a CNN-based spatial modeling method for Euclidean space. Another line of modeling methods using graph structures for managing complex road networks (e.g., GCNs) have also become popular. However, using GCNs requires building an adjacency matrix, and GCNs depend heavily on pre-defined graph structure.

To overcome these difficulties, several approaches, such as graph attention models, have been proposed for dynamic edge importance weighting (Park et al., 2020). Graph-WaveNet (Wu et al., 2019) uses a learnable static adjacency matrix to capture hidden spatial dependencies in training.

SLCNN (Zhang et al., 2020) and StemGNN (Cao et al., 2020) try to learn a time-varying graph by projecting current traffic conditions. MegaCRN (Jiang et al., 2023) uses memory-based graph learning to construct a noise-robust graph. Despite their effectiveness, forecasting models still suffer from inaccurate predictions due to abruptly changing speeds, instability, and changes in spatial dependency. To address these challenges, we design TESTAM to change its spatial modeling methods based on the traffic context using the Mixture-of-Experts technique.

## 2.2 MIXTURE OF EXPERTS

The Mixture-of-Experts (MoEs) is a machine learning technique devised by Shazeer et al. (2017) that has been actively researched as a powerful method for increasing model capacities without additional computational costs. MoEs have been used in various machine learning tasks, such as computer vision (Dryden & Hoefler, 2022) and natural language processing (Zhou et al., 2022; Fedus et al., 2022). Recently, MoEs have gone beyond being the purpose of increasing model capacities and are used to "specialize" each expert in subtasks at specific levels, such as the sample (Eigen et al., 2014; McGill & Perona, 2017; Rosenbaum et al., 2018), token (Shazeer et al., 2017; Fedus et al., 2022), and patch levels (Riquelme et al., 2021). These coarse-grained routing of the MoEs are frequently trained with multiple auxiliary losses, focusing on load balancing (Fedus et al., 2022; Dryden & Hoefler, 2022), but it often causes the experts to lose their opportunity to specialize. Furthermore, MoEs assign identical structures to every expert, eventually leading to limitations caused by the architecture, such as sharing the same inductive bias, which hardly changes. Dryden & Hoefler (2022) have proposed Spatial Mixture-of-Experts (SMoEs) that introduces fine-grained routing to solve the regression problem. SMOEs induce inductive bias via fine-grained, location-dependent routing for regression problems. They utilize one routing classification loss based on the final output losses, penalize gating networks with output error signals, and reduce the change caused by inaccurate routing for better routing and expert specialization. However, SMoEs only attempt to avoid incorrect routing and pay less attention to the best routing. TESTAM differs from existing MoEs in two main ways: it utilizes experts with different spatial modeling methods for better generalization, and it can be optimized with two loss functions–one for avoiding the worst route and another for choosing the best route for better specialization.

## 3 METHODS

### 3.1 PRELIMINARIES

**Problem Definition** Let us define a road network as $\mathcal{G} = (\mathcal{V}, \mathcal{E}, \mathcal{A})$, where $\mathcal{V}$ is a set of all roads in road networks with $|\mathcal{V}| = N$, $\mathcal{E}$ is a set of edges representing the connectivity between roads, and $\mathcal{A} \in \mathbb{R}^{N \times N}$ is a matrix representing the topology of $\mathcal{G}$. Given road networks, we formulate our problem as a special version of multivariate time series forecasting that predicts future $T$ graph signals based on $T'$ historical input graph signals:

$$\left[ X_{\mathcal{G}}^{(t-T'+1)}, \dots, X_{\mathcal{G}}^{(t)} \right] \xrightarrow{f(\cdot)} \left[ X_{\mathcal{G}}^{(t+1)}, \dots, X_{\mathcal{G}}^{(t+T)} \right],$$

where $X_{\mathcal{G}}^{(i)} \in \mathbb{R}^{N \times C}$, $C$ is the number of input features. We aim to train the mapping function $f(\cdot): \mathbb{R}^{T' \times N \times C} \to \mathbb{R}^{T \times N \times C}$, which predicts the next $T$ steps based on the given $T'$ observations. For the sake of simplicity, we omit $\mathcal{G}$ from $X_{\mathcal{G}}$ hereafter.

**Spatial Modeling Methods in Traffic Forecasting** To effectively forecast the traffic signals, we first discuss spatial modeling, which is one of the necessities for traffic data modeling. In traffic forecasting, we can classify spatial modeling methods into four categories: 1) with identity matrix (i.e., multivariate time-series forecasting), 2) with a pre-defined adjacency matrix, 3) with a trainable adjacency matrix, and 4) with attention (i.e., dynamic spatial modeling without prior knowledge). Conventionally, a graph topology $\mathcal{A}$ is constructed via an empirical law, including inverse distance (Li et al., 2018; Yu et al., 2018) and cosine similarity (Geng et al., 2019). However, these empirically built graph structures are not necessarily optimal, thus often resulting in poor spatial modeling quality. To address this challenge, a line of research (Wu et al., 2019; Bai et al., 2020; Jiang et al., 2023) is proposed to capture the hidden spatial information. Specifically, a trainable

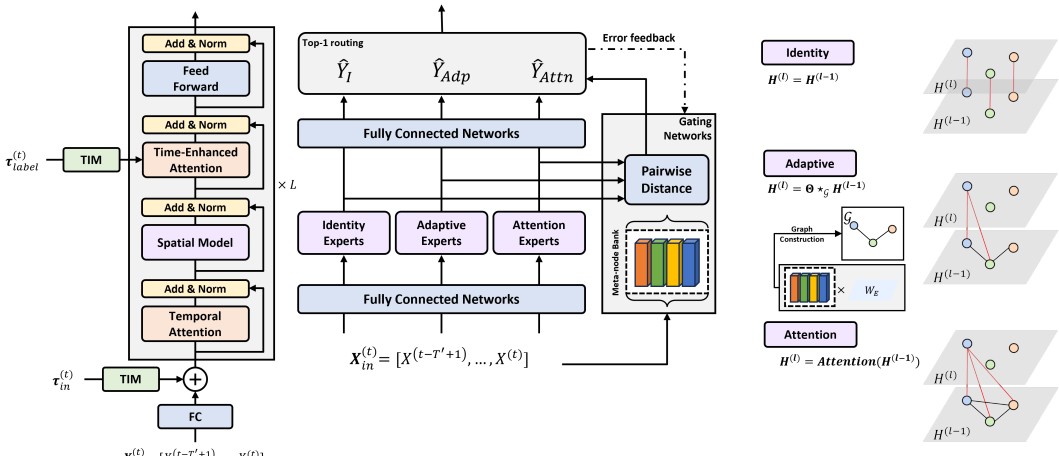

Figure 1: Overview of TESTAM. **Left**: The architecture of each expert. **Middle**: The workflow and routing mechanism of TESTAM. Solid lines indicate forward paths, and the dashed lines represent backward paths. **Right**: The three spatial modeling methods of TESTAM. The black lines indicate spatial connectivity, and red lines represent information flow corresponding to spatial connectivity. Identity, adaptive, and attention experts are responsible for temporal modeling, spatial modeling with learnable static graph, and with dynamic graph (i.e., attention), respectively.

function $g(\cdot, \theta)$ is used to derive the optimal topological representation $\tilde{\mathcal{A}}$ as:

$$\tilde{\mathcal{A}} = softmax(\text{relu}(g(X^{(t)}, \theta), g(X^{(t)}, \theta)^{\top})), \tag{1}$$

where $g(X^{(t)}, \theta) \in \mathbb{R}^{N \times e}$, and $e$ is the embedding size. Spatial modeling based on Eq. 1 can be classified into two subcategories according to whether $g(\cdot, \theta)$ depends on $X^{(t)}$. Wu et al. (2019) define $g(\cdot, \theta) = E \in \mathbb{R}^{N \times e}$, which is time-independent and less noise-sensitive but less in-situ modeling. Cao et al. (2020); Zhang et al. (2020) propose time-varying graph structure modeling with $g(H^{(t)}, \theta) = H^{(t)}W$, where $W \in \mathbb{R}^{d \times e}$, projecting hidden states to another embedding space. Ideally, this method models dynamic changes in graph topology, but it is noise-sensitive.

To reduce noise-sensitivity and obtain a time-varying graph structure, Zheng et al. (2020) adopt a spatial attention mechanism for traffic forecasting. Given input $H_i$ of node $i$ and its spatial neighbor $\mathcal{N}_i$, they compute spatial attention using multi-head attention as follows:

$$H_i^* = \text{Concat}(o_i^{(1)}, \dots, o_i^{(K)})W^O; \qquad o_i^{(k)} = \sum_{s \in \mathcal{N}_i} \alpha_{i,s} \cdot f_v^{(k)}(H_s) \tag{2}$$

$$\alpha_{i,j} = \frac{\exp(e_{i,j})}{\sum_{s \in \mathcal{N}_i} \exp(e_{i,s})}; \qquad e_{i,j} = \frac{\left(f_q^{(k)}(H_i)\right)\left(f_k^{(k)}(H_j)\right)^{\top}}{\sqrt{d_k}}, \tag{3}$$

where $W^O$ is a projection layer, $d_k$ is a dimension of key vector, and $f_q^{(k)}(\cdot)$, $f_k^{(k)}(\cdot)$, and $f_v^{(k)}(\cdot)$ are query, key, and value projections of the $k$-th head, respectively. Although effective, these attention-based approaches still suffer from irregular spatial modeling, such as less accurate self-attention (i.e., from node $i$ to $i$) (Park et al., 2020) and uniformly distributed uninformative attention, regardless of its spatial relationships (Jin et al., 2023).

## 3.2 MODEL ARCHITECTURE

Although transformers are well-established structures for time-series forecasting, it has a couple of problems when used for spatio-temporal modeling: they do not consider spatial modeling, consume considerable memory resources, and have bottleneck problems caused by the autoregressive decoding process. Park et al. (2020) have introduced an improved transformer model with graph attention (GAT), but the model still has auto-regressive properties. To eliminate the autoregressive characteristics while preserving the advantage of the encoder–decoder architecture, TESTAM transfers the

attention domain through time-enhanced attention and temporal information embedding. As shown in Fig. 1 (left), in addition to temporal information embedding, each expert layer consists of four sublayers: temporal attention, spatial modeling, time-enhanced attention, and point-wise feed-forward neural networks. Each sublayer is connected to a bypass through skip connections. To improve generalization, we apply layer normalization after each sublayer. All experts have the same hidden size and number of layers and differ only in terms of spatial modeling methods.

**Temporal Information Embedding** Since temporal features (e.g., time of day) work as a global position with a specific periodicity, we omit position embedding in the original transformer architecture. Furthermore, instead of normalized temporal features, we utilize Time2Vec embedding (Kazemi et al., 2019) for periodicity and linearity modeling. Specifically, for the temporal feature $\tau \in \mathbb{N}$, we represent $\tau$ with $h$-dimensional embedding vector $v(\tau)$ and the learnable parameters $w_i, \phi_i$ for each embedding dimension $i$ as below:

$$TIM(\tau)[i] = \begin{cases} w_i v(\tau)[i] + \phi_i, & \text{if } i = 0 \\ \mathcal{F}(w_i v(\tau)[i] + \phi_i) & \text{if } 1 \leq i \leq h-1, \end{cases} \qquad (4)$$

where $\mathcal{F}$ is a periodic activation function. Using Time2Vec embedding, we enable the model to utilize the temporal information of labels. Here, temporal information embedding of an input sequence is concatenated with other input features and then projected onto the hidden size $h$.

**Temporal Attention** As temporal attention in TESTAM is the same as that of transformers, we describe the benefits of temporal attention. Recent studies (Li et al., 2018; Bai et al., 2020) have shown that attention is an appealing solution for temporal modeling because, unlike recurrent unit-based or convolution-based temporal modeling, it can be used to directly attend to features across time steps with no restrictions. Temporal attention allows parallel computation and is beneficial for long-term sequence modeling. Moreover, it has less inductive bias in terms of locality and sequentiality. Although strong inductive bias can help the training, less inductive bias enables better generalization. Furthermore, for the traffic forecasting problem, causality among roads is an unavoidable factor (Jin et al., 2023) that cannot be easily modeled in the presence of strong inductive bias, such as sequentiality or locality.

**Spatial Modeling Layer** In this work, we leverage three spatial modeling layers for each expert, as shown in the middle of Fig. 1: spatial modeling with an identity matrix (i.e., no spatial modeling), spatial modeling with a learnable adjacency matrix (Eq. 1), and spatial modeling with attention (Eq. 2 and Eq. 3). We calculate spatial attention using Eqs. 2 and 3. Specifically, we compute attention with $\forall_{i \in \mathcal{V}}, \mathcal{N}_i = \mathcal{V}$, which means attention with no spatial restrictions. This setting enables similarity-based attention, resulting in better generalization.

Inspired by the success of memory-augmented graph structure learning (Jiang et al., 2023; Lee et al., 2022), we propose a modified meta-graph learner that learns prototypes from both spatial graph modeling and gating networks. Our meta-graph learner consists of two individual neural networks with a meta-node bank $\mathbf{M} \in \mathbb{R}^{m \times e}$, where $m$ and $e$ denote total memory items and a dimension of each memory, respectively, a hyper-network (Ha et al., 2017) for generating node embedding conditioned on $\mathbf{M}$, and gating networks to calculate the similarities between experts' hidden states and queried memory items. In this section, we mainly focus on the hyper-network. We construct a graph structure with a meta-node bank $\mathbf{M}$ and a projection $W_E \in \mathbb{R}^{e \times d}$ as follows:

$$E = \mathbf{M}W_E; \tilde{A} = softmax(\mathsf{relu}(EE^\top))$$

By constructing a memory-augmented graph, the model achieves better context-aware spatial modeling than that achieved using other learnable static graphs (e.g., graph modeling with $E \in \mathbb{R}^{N \times d}$). Detailed explanations for end-to-end training and meta-node bank queries are provided in Sec. 3.3.

**Time-Enhanced Attention** To eliminate the error propagation effects caused by auto-regressive characteristics, we propose a time-enhanced attention layer that helps the model transfer its domain from historical $T'$ time steps (i.e., source domain) to next $T$ time steps (i.e., target domain). Let

$\tau_{label}^{(t)} = [\tau^{(t+1)}, \ldots, \tau^{(t+T)}]$ be a temporal feature vector of the label. We calculate the attention score from the source time step $i$ to the target time step $j$ as:

$$\alpha_{i,j} = \frac{\exp(e_{i,j})}{\sum_{k=t+1}^{T} \exp(e_{i,k})},$$

$$e_{i,j} = \frac{(H^{(i)}W_q^{(k)})(\text{TIM}(\tau^{(j)})W_k^{(k)})^\top}{\sqrt{d_k}}, \tag{5}$$

where $d_k = d/K$, $K$ is the number of heads, and $W_q^{(k)}, W_k^{(k)}$ are linear transformation matrices. We can calculate the attention output using the same process as in Eq. 2, except that time-enhanced attention attends to the time steps of each node, whereas Eq. 2 attends to the important nodes at each time step.

## 3.3 GATING NETWORKS

In this section, we describe the gating networks used for in-situ routing. Conventional MoE models have multiple experts with the same architecture and conduct coarse-grained routing, focusing on increasing model capacity without additional computational costs (Shazeer et al., 2017). However, coarse-grained routing provides experts with limited opportunities for specialization. Furthermore, in the case of the regression problem, existing MoEs hardly change their routing decisions after initialization because the gate is not guided by the gradients of regression tasks, as Dryden & Hoefler (2022) have revealed. Consequently, gating networks cause "mismatches," resulting in uninformative and unchanging routing. Moreover, using the same architecture for all experts is less beneficial in terms of generalization since they also share the same inductive bias.

To resolve this issue, we propose novel memory-based gating networks and two classification losses with regression error-based pseudo labels. Existing memory-based traffic forecasting approaches (Lee et al., 2022; Jiang et al., 2023) reconstruct the encoder's hidden state with memory items, allowing memory to store typical features from seen samples for pattern matching. In contrast, we aim to learn the direct relationship between input signals and output representations. For node $i$ at time step $t$, we define the memory-querying process as follows:

$$Q_i^{(t)} = X_i^{(t)}W_q + b_q$$

$$\begin{cases} a_j = \frac{\exp(Q_i^{(t)}M[j]^\top)}{\sum_{j=1}^{m} \exp(Q_i^{(t)}M[j]^\top)} \\ O_i^{(t)} = \sum_{j=1}^{m} a_j M[j] \end{cases},$$

where $M[i]$ is the $i$-th memory item, and $W_q$ and $b_q$ are learnable parameters for input projection. Let $z_e$ be an output representation of expert $e$. Given the queried memory $O_i^{(t)} \in \mathbb{R}^e$, we calculate the routing probability $p_e$ as shown below:

$$r_e = g(z_e, O_i^{(t)}); \quad p_e = \frac{r_e}{\sum_{e \in [e_1, \ldots, e_E]} r_e},$$

where $E$ is the number of experts. Since we use the similarity between output states and queried memory as the routing probability, solving the routing problem induces memory learning of a typical output representation and input-output relationship. We select the top-1 expert output as final output.

**Routing Classification Losses** To enable fine-grained routing that fits the regression problem, we adopt two classification losses: a classification loss to avoid the worst routing and another loss function to find the best routing. Inspired by SMoEs, we define the worst routing avoidance loss as the cross entropy loss with pseudo label $l_e$ as shown below:

$$L_{worst}(\mathbf{p}) = -\frac{1}{E} \sum_e l_e log(p_e) \tag{6}$$

$$l_e = \begin{cases} 1 & \text{if } L(y, \hat{y}) \text{ is smaller than } q\text{-th quantile and } p_e = argmax(\mathbf{p}) \\ 1/(E-1) & \text{if } L(y, \hat{y}) \text{ is greater than } q\text{-th quantile and } p_e \neq argmax(\mathbf{p}) \\ 0 & otherwise \end{cases},$$

Table 1: Experimental results on three real-world datasets with 13 baseline models and TESTAM. The values in bold indicate the best, and underlined values indicate the second-best performance.

| METR-LA | 15 min | | | 30 min | | | 60 min | | |
|---|---|---|---|---|---|---|---|---|---|
| | MAE | RMSE | MAPE | MAE | RMSE | MAPE | MAE | RMSE | MAPE |
| HA (Li et al., 2018) | 4.16 | 7.80 | 13.00% | 4.16 | 7.80 | 13.00% | 4.16 | 7.80 | 13.00% |
| STGCN (Yu et al., 2018) | 2.88 | 5.74 | 7.62% | 3.47 | 7.24 | 9.57% | 4.59 | 9.40 | 12.70% |
| DCRNN (Li et al., 2018) | 2.77 | 5.38 | 7.30% | 3.15 | 6.45 | 8.80% | 3.60 | 7.59 | 10.50% |
| Graph-WaveNet (Wu et al., 2019) | 2.69 | 5.15 | 6.90% | 3.07 | 6.22 | 8.37% | 3.53 | 7.37 | 10.01% |
| STTN (Xu et al., 2020) | 2.79 | 5.48 | 7.19% | 3.16 | 6.50 | 8.53% | 3.60 | 7.60 | 10.16% |
| GMAN (Zheng et al., 2020) | 2.80 | 5.55 | 7.41% | 3.12 | 6.49 | 8.73% | 3.44 | 7.35 | 10.07% |
| MTGNN (Wu et al., 2020) | 2.69 | 5.18 | 6.86% | 3.05 | 6.17 | 8.19% | 3.49 | 7.23 | 9.87% |
| StemGNN (Cao et al., 2020) | 2.56 | 5.06 | 6.46% | 3.01 | 6.03 | 8.23% | 3.43 | 7.23 | 9.85% |
| AGCRN (Bai et al., 2020) | 2.86 | 5.55 | 7.55% | 3.25 | 6.57 | 8.99% | 3.68 | 7.56 | 10.46% |
| CCRNN (Ye et al., 2021) | 2.85 | 5.54 | 7.50% | 3.24 | 6.54 | 8.90% | 3.73 | 7.65 | 10.59% |
| GTS (Shang et al., 2021) | 2.65 | 5.20 | 6.80% | 3.05 | 6.22 | 8.28% | 3.47 | 7.29 | 9.83% |
| PM-MemNet (Lee et al., 2022) | 2.65 | 5.29 | 7.01% | 3.03 | 6.29 | 8.42% | 3.46 | 7.29 | 9.97% |
| MegaCRN (Jiang et al., 2023) | **2.52** | 4.94 | 6.44% | **2.93** | 6.06 | 7.96% | 3.38 | 7.23 | 9.72% |
| TESTAM | 2.54 | **4.93** | **6.42%** | 2.96 | **6.04** | **7.92%** | **3.36** | **7.09** | **9.67%** |

| PEMS-BAY | 15 min | | | 30 min | | | 60 min | | |
|---|---|---|---|---|---|---|---|---|---|
| | MAE | RMSE | MAPE | MAE | RMSE | MAPE | MAE | RMSE | MAPE |
| HA (Li et al., 2018) | 2.88 | 5.59 | 6.80% | 2.88 | 5.59 | 6.80% | 2.88 | 5.59 | 6.80% |
| STGCN (Yu et al., 2018) | 1.36 | 2.96 | 2.90% | 1.81 | 4.27 | 4.17% | 2.49 | 5.69 | 5.79% |
| DCRNN (Li et al., 2018) | 1.38 | 2.95 | 2.90% | 1.74 | 3.97 | 3.90% | 2.07 | 4.74 | 4.90% |
| Graph-WaveNet (Wu et al., 2019) | 1.30 | 2.74 | 2.73% | 1.63 | 3.70 | 3.67% | 1.95 | 4.52 | 4.63% |
| STTN (Xu et al., 2020) | 1.36 | 2.87 | 2.89% | 1.67 | 3.79 | 3.78% | 1.95 | 4.50 | 4.58% |
| GMAN (Zheng et al., 2020) | 1.35 | 2.90 | 2.87% | 1.65 | 3.82 | 3.74% | 1.92 | 4.49 | 4.52% |
| MTGNN (Wu et al., 2020) | 1.32 | 2.79 | 2.77% | 1.65 | 3.74 | 3.69% | 1.94 | 4.49 | 4.53% |
| StemGNN (Cao et al., 2020) | 1.23 | 2.48 | 2.63% | N/A from (Cao et al., 2020) | | | N/A from (Cao et al., 2020) | | |
| AGCRN (Bai et al., 2020) | 1.36 | 2.88 | 2.93% | 1.69 | 3.87 | 3.86% | 1.98 | 4.59 | 4.63% |
| CCRNN (Ye et al., 2021) | 1.38 | 2.90 | 2.90% | 1.74 | 3.87 | 3.90% | 2.07 | 4.65 | 4.87% |
| GTS (Shang et al., 2021) | 1.34 | 2.84 | 2.83% | 1.67 | 3.83 | 3.79% | 1.98 | 4.56 | 4.59% |
| PM-MemNet (Lee et al., 2022) | 1.34 | 2.82 | 2.81% | 1.65 | 3.76 | 3.71% | 1.95 | 4.49 | 4.54% |
| MegaCRN (Jiang et al., 2023) | **1.28** | **2.72** | 2.67% | 1.60 | 3.68 | 3.57% | 1.88 | 4.42 | 4.41% |
| TESTAM | 1.29 | 2.77 | **2.61%** | **1.59** | **3.65** | **3.56%** | **1.85** | **4.33** | **4.31%** |

| EXPY-TKY | 10 min | | | 30 min | | | 60 min | | |
|---|---|---|---|---|---|---|---|---|---|
| | MAE | RMSE | MAPE | MAE | RMSE | MAPE | MAE | RMSE | MAPE |
| HA (Li et al., 2018) | 7.63 | 11.96 | 31.26% | 7.63 | 11.96 | 31.25% | 7.63 | 11.96 | 31.24% |
| STGCN (Yu et al., 2018) | 6.09 | 9.60 | 24.84% | 6.91 | 10.99 | 30.24% | 8.41 | 12.70 | 32.90% |
| DCRNN (Li et al., 2018) | 6.04 | 9.44 | 25.54% | 6.85 | 10.87 | 31.02% | 7.45 | 11.86 | 34.61% |
| Graph-WaveNet (Wu et al., 2019) | 5.91 | 9.30 | 25.22% | 6.59 | 10.54 | 29.78% | 6.89 | 11.07 | 31.71% |
| STTN (Xu et al., 2020) | 5.90 | 9.27 | 25.67% | 6.53 | 10.40 | 29.82% | 6.99 | 11.23 | 32.52% |
| GMAN (Zheng et al., 2020) | 6.09 | 9.49 | 26.52% | 6.64 | 10.55 | 30.19% | 7.05 | 11.28 | 32.91% |
| MTGNN (Wu et al., 2020) | 5.86 | 9.26 | 24.80% | 6.49 | 10.44 | 29.23% | 6.81 | 11.01 | 31.39% |
| StemGNN (Cao et al., 2020) | 6.08 | 9.46 | 25.87% | 6.85 | 10.80 | 31.25% | 7.46 | 11.88 | 35.31% |
| AGCRN (Bai et al., 2020) | 5.99 | 9.38 | 25.71% | 6.64 | 10.63 | 29.81% | 6.99 | 11.29 | 32.13% |
| CCRNN (Ye et al., 2021) | 5.90 | 9.29 | 24.53% | 6.68 | 10.77 | 29.93% | 7.11 | 11.56 | 32.56% |
| GTS (Shang et al., 2021) | - | - | - | - | - | - | - | - | - |
| PM-MemNet (Lee et al., 2022) | 5.94 | 9.25 | 25.10% | 6.52 | 10.42 | 29.00% | 6.87 | 11.14 | 31.22% |
| MegaCRN (Jiang et al., 2023) | **5.81** | **9.20** | **24.49%** | 6.44 | 10.33 | 28.92% | 6.83 | 11.04 | 31.02% |
| TESTAM | 5.84 | 9.23 | 25.36% | **6.42** | **10.24** | **28.90%** | **6.75** | **11.01** | **31.01%** |

where $\hat{y}$ is the output of the selected expert, and $q$ is an error quantile. If an expert is incorrectly selected, its label becomes zero and the unselected experts have the pseudo label $1/(E-1)$, which means that there are equal chances of choosing unselected experts.

We also propose the best-route selection loss for more precise routing. However, as traffic data are noisy and contain many nonstationary characteristics, the best-route selection is not an easy task. Therefore, instead of choosing the best routing for every time step and every node, we calculate node-wise routing. Our best-route selection loss is similar to that in Eq. 6, except that it calculates node-wise pseudo labels and the routing probability, and the condition for pseudo labels is changed from "$L(y, \hat{y})$ is greater/smaller than $q$-th quantile" to "$L(y, \hat{y})$ is greater/smaller than $(1 - q)$-th quantile." Detailed explanations are provided in Appendix A.

## 4 EXPERIMENTS

In this section, we describe experiments and compare the accuracy of TESTAM with that of existing models. We use three benchmark datasets for the experiments: METR-LA, PEMS-BAY, and EXPY-TKY. METR-LA and PEMS-BAY contain four-month speed data recorded by 207 sensors on Los Angeles highways and 325 sensors on Bay Area, respectively (Li et al., 2018). EXPY-TKY consists of three-month speed data collected from 1843 links in Tokyo, Japan. As EXPY-TKY covers a larger number of roads in a smaller area, its spatial dependencies with many abruptly changing speed patterns are more difficult to model than those in METR-LA or PEMS-BAY. METR-LA and PEMS-BAY datasets have 5-minute interval speeds and timestamps, whereas EXPY-TKY has 10-minute interval speeds and timestamps. Before training TESTAM, we have performed z-score normalization. In the cases of METR-LA and PEMS-BAY, we use 70% of the data for training, 10% for validation, and 20% for evaluation. For the EXPY-TKY, we utilize the first two months for training and validation and the last month for testing, as in the MegaCRN paper (Jiang et al., 2023).

## 4.1 EXPERIMENTAL SETTINGS

For all three datasets, we initialize the parameters and embedding using Xavier initialization. After performing a greedy search for hyperparameters, we set the hidden size $d = e = 32$, the memory size $m = 20$, the number of layers $l = 3$, the number of heads $K = 4$, the hidden size for the feed-forward networks $h_{ff} = 128$, and the error quantile $q = 0.7$. We use the Adam optimizer with $\beta_1 = 0.9, \beta_2 = 0.98$, and $\epsilon = 10^{-9}$, as in Vaswani et al. (2017). We vary the learning rate during training using the cosine annealing warmup restart scheduler (Loshchilov & Hutter, 2017) according to the formula below:

$$lrate = \begin{cases} lr_{min} + (lr_{max} - lr_{min}) \cdot \frac{T_{cur}}{T_{warm}} & \text{For the first } T_{warm} \text{ steps} \\ lr_{min} + \frac{1}{2}(lr_{max} - lr_{min})\big(1 + cos(\frac{T_{cur}}{T_{freq}}\pi)\big) & \text{otherwise} \end{cases}, \quad (7)$$

where $T_{cur}$ is the number of steps since the last restart. We use $T_{warm} = T_{freq} = 4000, lr_{min} = 10^{-7}$ for all datasets and set $lr_{max} = 3 * 10^{-3}$ for METR-LA and PEMS-BAY and $lr_{max} = 3 * 10^{-4}$ for EXPY-TKY. We follow the traditional 12-sequence (1 hour) input and 12-sequence output forecasting setting for METR-LA and PEMS-BAY and the 6-sequence (1 hour) input and 6-sequence output setting for EXPY-TKY, as in Jiang et al. (2023). We utilize mean absolute error (MAE) as a loss function and root mean squared error (RMSE) and mean absolute percentage error (MAPE) as evaluation metrics. All experiments are conducted using an RTX 3090 GPU.

We compare TESTAM with 13 baseline models: (1) historical average; (2) STGCN (Yu et al., 2018), a model with GCNs and CNNs; (3) DCRNN (Li et al., 2018), a model with graph convolutional recurrent units; (4) Graph-WaveNet (Wu et al., 2019) with a parameterized adjacency matrix; (5) STTN (Xu et al., 2020) and (6) GMAN (Zheng et al., 2020), state-of-the-art attention-based models; (7) MTGNN (Wu et al., 2020), (8) StemGNN (Cao et al., 2020), and (9) AGCRN (Bai et al., 2020), advanced models with an adaptive matrix; (10) CCRNN (Ye et al., 2021), a model with multiple adaptive matrices; (11) GTS (Shang et al., 2021), a model with a graph constructed with long-term historical data; and (12) PM-MemNet (Lee et al., 2022) and (13) MegaCRN (Jiang et al., 2023), state-of-the-art models with memory units.

## 4.2 EXPERIMENTAL RESULTS

The experimental results are shown in Table 1. TESTAM outperforms all other models, especially in long-term predictions, which are usually more difficult. Note that we use the results reported in the respecive papers after comparing them with reproduced results from official codes provided by the authors. The models with learnable static graphs (Graph-WaveNet, MTGNN, and CCRNN) and dynamic graphs (STTN and GMAN) show competitive performance, indicating that they have certain advantages. In terms of temporal modeling, RNN-based temporal models (DCRNN and AGCRN) show worse performance than the other methods in long-term forecasting due to error-accumulation of RNNs. Conversely, MegaCRN and PM-MemNet maintained their advantages even in long-term forecasting by injecting a memory-augmented representation vector into the decoder. GMAN and StemGNN have performed worse with EXPY-TKY, indicating a disadvantage of the attention methods, such as long-tail problems and uniformly distributed attention (Jin et al., 2023).

As EXPY-TKY has a 6–9 times larger number of roads than the other two datasets, experimental results with EXPY-TKY highlight the importance of spatial modeling. For example, attention-based spatial modeling methods show disadvantages and the results of modeling with time-varying networks (e.g., StemGNN) suggest that it could not properly capture spatial dependencies. In contrast, our model, TESTAM, shows its superiority to all other models, including those with learnable matrices. The results demonstrate that in-situ spatial modeling is crucial for traffic forecasting.

## 4.3 ABLATION STUDY

The ablation study has two goals: to evaluate actual improvements achieved by each method, and to test two hypotheses: (1) in-situ modeling with diverse graph structures is advantageous for traffic forecasting and (2) having two loss functions for avoiding the worst route and leading to the best route is effective. To achieve these aims, we have designed a set of TESTAM variants, which are described below:

Table 2: Ablation study results across all prediction windows (i.e., average performance)

| Ablation | METR-LA | | | PEMS-BAY | | | EXPY-TKY | | |
|---|---|---|---|---|---|---|---|---|---|
| | MAE | RMSE | MAPE | MAE | RMSE | MAPE | MAE | RMSE | MAPE |
| w/o gating | 3.00 | 6.12 | 8.29% | 1.58 | 3.57 | 3.53% | 6.74 | 10.97 | 29.48% |
| Ensemble | 2.98 | 6.08 | 8.12% | 1.56 | 3.53 | 3.50% | 6.66 | 10.68 | 29.43% |
| worst-route avoidance only | 2.96 | 6.06 | 8.11% | 1.55 | 3.52 | 3.48% | 6.45 | 10.50 | 28.70% |
| Replaced | 2.97 | 6.04 | 8.05% | 1.56 | 3.54 | 3.47% | 6.56 | 10.62 | 29.20% |
| w/o TIM | 2.96 | 5.98 | 8.07% | 1.54 | 3.45 | 3.46% | 6.44 | 10.40 | 28.94% |
| w/o time-enhanced attention | 2.99 | 6.03 | 8.15% | 1.58 | 3.59 | 3.52% | 6.64 | 10.75 | 29.85% |
| TESTAM | **2.93** | **5.95** | **7.99%** | **1.53** | **3.47** | **3.41%** | **6.40** | **10.40** | **28.67%** |

**w/o gating**    It uses only the output of the attention experts without ensembles or any other gating mechanism. Memory items are not trained because there are no gradient flows for the adaptive expert or gating networks. This setting results in an architecture similar to that of GMAN.

**Ensemble**    Instead of using MoEs, the final output is calculated with the weighted summation of the gating networks and each expert's output. This setting allows the use of all spatial modeling methods but no in-situ modeling.

**worst-route avoidance only**    It excludes the loss for guiding best route selection. The exclusion of this relatively coarse-grained loss function is based on the fact that coarse-grained routing tends not to change its decisions after initialization (Dryden & Hoefler, 2022).

**Replaced**    It does not exclude any components. Instead, it replaces identity expert with a GCN-based adaptive expert, reducing spatial modeling diversity. The purpose of this setting is to test the hypothesis that in-situ modeling with diverse graph structures is helpful for traffic forecasting.

**w/o TIM**    It replaces temporal information embedding (TIM) with simple embedding vectors without periodic activation functions.

**w/o time-enhanced attention**    It replaces time-enhanced attention with basic temporal attention as we described in Sec. 3.2.

The experimental results shown in Table 2 connote that our hypotheses are supported and that TESTAM is a complete and indivisible set. The results of "w/o gating" and "ensemble" suggest that in-situ modeling greatly improves the traffic forecasting quality. The "w/o gating" results indicate that the performance improvement is not due to our model but due to in-situ modeling itself since this setting lead to performance comparable to that of GMAN (Zheng et al., 2020). "worst-route avoidance only" results indicate that our hypothesis that both of our routing classification losses are crucial for proper routing is valid. Finally, the results of "replaced," which indicate significantly worse performance even than "worst route avoidance only," confirm the hypothesis that diverse graph structures is helpful for in-situ modeling. Additional qualitative results with examples are provided in Appendix C.

## 5    CONCLUSION

In this paper, we propose the time-enhanced spatio-temporal attention model (TESTAM), a novel Mixture-of-Experts model with attention that enables effective in-situ spatial modeling in both recurring and non-recurring situations. By transforming a routing problem into a classification task, TESTAM can contextualize various traffic conditions and choose the most appropriate spatial modeling method. TESTAM achieves superior performance to that of existing traffic forecasting models in three real-world datasets: METR-LA, PEMS-BAY, and EXPY-TKY. The results obtained using the EXPY-TKY dataset indicate that TESTAM is highly advantageous for large-scale graph structures, which are more applicable to real-world problems. We have also obtained qualitative results to visualize when and where TESTAM chooses specific graph structures. In future work, we plan to further improve and generalize TESTAM for the other spatio-temporal and multivariate time series forecasting tasks.

ACKNOWLEDGMENTS

This work was supported by the National Research Foundation of Korea (NRF) grant funded by the Korea government (MSIT) (No.RS-2023-00218913, No. 2021R1A2C1004542), by the Institute of Information & Communications Technology Planning & Evaluation (IITP) grants (No. 2020-0-01336–Artificial Intelligence Graduate School Program, UNIST), and by the Green Venture R&D Program (No. S3236472), funded by the Ministry of SMEs and Startups (MSS, Korea)

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

## A    ROUTING CLASSIFICATION LOSS FUNCTION

In this section, we provide detailed information on the routing classification loss function. Both functions for worst-route avoidance and best-route selection are cross-entropy loss functions with different pseudo labels and routing levels. For the worst-route avoidance, we compute fine-grained routing for each point of each road, as Dryden & Hoefler (2022) do. However, utilizing worst-route avoidance is suboptimal because experts have less opportunities to be specialized for the best routing. Therefore, we adopt the best-route selection loss function for the routing problem. While designing the best-route selection loss function, we have two main concerns: 1) traffic data often shows severe fluctuation, which prevents a model to consistently choose best-fit experts, and 2) the best-route selection itself is a more complex task than worst-route avoidance, resulting in the model being hardly trained with fine-grained routing. To overcome those challenges, we have decided to construct node-wise best-route selection loss.

### A.1    WORST-ROUTE AVOIDANCE LOSS

For the worst-route avoidance loss function, we have built our pseudo label $l_e$ as Eq. 6. In this section, we describe how those labels are chosen. Given prediction $\hat{Y} \in \mathbb{R}^{N \times T}$ and ground truth $Y \in \mathbb{R}^{N \times T}$, we have point-wise distance $L(Y, \hat{Y}) \in \mathbb{R}^{N \times T}$ between prediction and ground truth. Given point-wise distances and error quantile $q$, we say that routing of road $n$ at time $t$ is incorrect (or the worst routing) if $L(y_{n,t}, \hat{y}_{n,t})$ is greater than $q$-th quantile. Therefore, if $L(y_{n,t}, \hat{y}_{n,t})$ is greater than $q$-th quantile, the pseudo label of the selected expert will be zero, and the labels for the other unselected experts will be $1/(E-1)$, where $E$ is total number of experts. However, if $L(y_{n,t}, \hat{y}_{n,t})$ is smaller than $q$-th quantile, which means it is correctly routed and the worst route is avoided, the selected experts will have a pseudo label of one and the other experts will have a pseudo label of zero. Formally, we can define pseudo label of expert $e$ as follows:

$$l_e = \begin{cases} 1 & \text{if } L(y, \hat{y}) \text{ is smaller than } q\text{-th quantile and } p_e = argmax(\mathbf{p}) \\ 1/(E-1) & \text{if } L(y, \hat{y}) \text{ is greater than } q\text{-th quantile and } p_e \neq argmax(\mathbf{p}) \\ 0 & otherwise \end{cases}$$

### A.2    BEST-ROUTE SELECTION LOSS

For the best-route selection loss function, we define node-wise pseudo labels by converting each condition of pseudo labeling for worst-route avoidance. For worst-route avoidance, we assume that the routing is incorrect (i.e., the worst) if $L(y_{n,t}\hat{y}_{n,t})$ is greater than $q$-th quantile. In the best-route selection, we define that the routing is correct (i.e., best) if $L(y_n\hat{y}_n)$ is smaller than $1-q$-th quantile; otherwise, its incorrectly routed, as shown below:

$$l_e = \begin{cases} 1 & \text{if } L(y, \hat{y}) \text{ is smaller than } 1-q\text{-th quantile and } p_e = argmax(\mathbf{p}) \\ 1/(E-1) & \text{if } L(y, \hat{y}) \text{ is greater than } 1-q\text{-th quantile and } p_e \neq argmax(\mathbf{p}) \\ 0 & otherwise \end{cases}$$

Table 3: Computation time of the models with the METR-LA dataset

|  | Training time/epoch | Inference time | # of params |
|---|---|---|---|
| STGCN | 14.8 secs | 16.70 secs | 320k |
| DCRNN | 122.22 secs | 13.44 secs | 372k |
| Graph-WaveNet | 48.07 secs | 3.69 secs | 309k |
| GMAN | 312.1 secs | 33.7 secs | 901k |
| MegaCRN | 84.7 secs | 11.76 secs | 339k |
| TESTAM | 150 secs | 7.96 secs | 224k |

## B  COMPUTATIONAL COST ANALYSIS

For the computational cost analysis, we use five models as baselines: 1) STGCN (Yu et al., 2018), the lightest model that utilizes GCNs and CNNs to forecast 1-step future traffic condition; 2) DCRNN (Li et al., 2018), a well-known traffic forecasting model with graph-convolutional recurrent units; 3) Graph-WaveNet (Wu et al., 2019), a model that forecasts values by parallel computation with GCNs and CNNs; 4) GMAN (Zheng et al., 2020), a spatio-temporal attention model for traffic forecasting, and 5) MegaCRN (Jiang et al., 2023), one of state-of-the-art models using GCRNN and memory network concepts.

We have investigated other models for comparison but decided to exclude them after careful considerations. For example, we have excluded MTGNN and StemGNN since they are an improved version of Graph-WaveNet and have similar computational costs compared to Graph-WaveNet. Similarly, AGCRN, CCRNN, and GTS are excluded from baselines because they are variants of DCRNN, with few changes in computational costs. PM-MemNet and MegaCRN utilize sequence-to-sequence modeling with shared memory units; however, PM-MemNet experiences computational bottleneck with its stacked memory units, which requires $L$ times larger computational costs than those of MegaCRN.

Even though TESTAM utilizes three individual experts for the prediction, we emphasize that it has a smaller number of parameters compared to the other models due to its small number of layers per each expert, which highly affects the computational costs. Furthermore, TESEAM only uses the encoder architecture of the transformer with a time-enhanced attention module that enables parallel computation, eliminating a computational bottleneck caused by the decoding process. As a result, in terms of computational costs, TESTAM is two times cheaper than the attention-based model (i.e., GMAN), illustrating a similar training time with DCRNN. Furthermore, in the inference phase, TESTAM shows the second fastest computation with the smallest number of the parameters.

## C  DETAILED EXPERIMENTAL RESULTS

Table 4 presents experimental results under various environment settings. In the table, we pose three scenarios: (I), (H), and (E). Each scenario represents difficult conditions in making accurate forecasting. (I) is a set of isolated roads, chosen by considering spatial locations and quantitative analysis results on the adjacency matrix. (H) is the hard-to-predict roads, including intersections and the roads with high traffic fluctuations. We have determined roads for (H) by visually exploring the roads (for intersections) and by selecting the roads with top 10% entropy-based time-series complexity. (E) contains roads and time with sudden events, including accidents, traffic controls, or holidays (e.g., Christmas). In the experiments, we compare the performances of three baselines with TESTAM, which are selected as a representative model for each spatial modeling method.

As shown in Table 4, TESTAM outperforms other models in non-recurring situations (i.e., (E)) and the roads that have unique spatial and topological features ((I) and (E)). Especially, TESTAM consistently proves its superiority for the roads with spatially unique features, outperforming existing models from 4% to 7% in general. Among all of three baselines, we observe that the attention-based modeling method has better encoded spatial information for most of the hard-to-predict scenarios. However, there are cases where the attention-based model fails, such as PEMS-BAY (I) or PEMS-BAY (H). From the perspective of temporal modeling, the attention-based modeling method indicates better long-term forecasting performance, while CNN- and RNN-based modeling methods are advantageous in short-term forecasting. TESTAM, in contrast, despite of its temporal modeling methods, it outperforms all the baselines in terms of both short-term and long-term forecasting cases. The results indicate that temporal information embedding and time-enhanced attention help the model to effectively transfer information from the input domain to the output domain.

Table 4: Case-specific experimental results on three real-world datasets. The numbers in bold mean the best performance, and those underlined mean the second-best performance. (I) means isolated roads. (H) means hard-to-predict roads, including intersections and the roads with high traffic fluctuations. (E) means the non-recurring circumstances, such as holidays or accidents.

| METR-LA (I) | 15 min | | | 30 min | | | 60 min | | |
|---|---|---|---|---|---|---|---|---|---|
| | MAE | RMSE | MAPE | MAE | RMSE | MAPE | MAE | RMSE | MAPE |
| Graph-WaveNet (Wu et al., 2019) | 3.58 | 5.93 | 8.43% | 3.90 | 6.74 | 9.59% | 4.29 | 7.50 | 10.98% |
| GMAN (Zheng et al., 2020) | 3.81 | 6.99 | 9.15% | 4.03 | 7.48 | 9.97% | 4.32 | 8.13 | 11.13% |
| MegaCRN (Jiang et al., 2023) | 3.54 | **5.88** | 8.46% | 3.88 | 6.69 | 9.74% | 4.35 | 7.67 | 11.39% |
| TESTAM | **3.52** | _5.89_ | **8.37%** | **3.80** | **6.59** | **9.43%** | **4.13** | **7.31** | **10.72%** |

| METR-LA (H) | 15 min | | | 30 min | | | 60 min | | |
|---|---|---|---|---|---|---|---|---|---|
| | MAE | RMSE | MAPE | MAE | RMSE | MAPE | MAE | RMSE | MAPE |
| Graph-WaveNet (Wu et al., 2019) | 4.07 | 6.74 | 11.75% | 4.73 | 8.03 | 14.38% | 5.48 | 9.34 | 17.20% |
| GMAN (Zheng et al., 2020) | 4.37 | 7.79 | 12.83% | 4.86 | 8.75 | 14.77% | 5.37 | 9.64 | 16.77% |
| MegaCRN (Jiang et al., 2023) | 4.02 | 6.68 | 11.61% | 4.73 | 8.13 | 14.46% | 5.55 | 9.72 | 17.67% |
| TESTAM | **3.96** | **6.62** | **11.42%** | **4.51** | **7.75** | **13.57%** | **5.19** | **9.03** | **16.04%** |

| METR-LA (E) | 15 min | | | 30 min | | | 60 min | | |
|---|---|---|---|---|---|---|---|---|---|
| | MAE | RMSE | MAPE | MAE | RMSE | MAPE | MAE | RMSE | MAPE |
| Graph-WaveNet (Wu et al., 2019) | 4.22 | 7.14 | 13.08% | 5.12 | 8.84 | 16.81% | 6.17 | 10.62 | 21.19% |
| GMAN (Zheng et al., 2020) | 4.45 | 7.67 | 14.49% | 5.16 | 9.13 | 17.53% | 5.95 | 10.58 | 21.18% |
| MegaCRN (Jiang et al., 2023) | **4.03** | **6.91** | **12.37%** | 4.96 | 8.75 | 16.10% | 6.01 | 10.69 | 20.58% |
| TESTAM | _4.11_ | _7.09_ | _12.54%_ | **4.92** | **8.71** | **15.71%** | **5.89** | **10.46** | **19.69%** |

| PEMS-BAY (I) | 15 min | | | 30 min | | | 60 min | | |
|---|---|---|---|---|---|---|---|---|---|
| | MAE | RMSE | MAPE | MAE | RMSE | MAPE | MAE | RMSE | MAPE |
| Graph-WaveNet (Wu et al., 2019) | 2.28 | 4.06 | 4.89% | 2.87 | 5.29 | 6.65% | 3.48 | 6.47 | 8.86% |
| GMAN (Zheng et al., 2020) | 2.51 | 5.12 | 5.65% | 3.06 | 6.20 | 7.34% | 3.55 | 7.10 | 8.90% |
| MegaCRN (Jiang et al., 2023) | 2.28 | 4.10 | 5.06% | 2.92 | 5.58 | 7.27% | 3.49 | 6.76 | 9.22% |
| TESTAM | **2.26** | **4.03** | **4.67%** | **2.86** | **5.24** | **6.45%** | **3.36** | **6.45** | **8.55%** |

| PEMS-BAY (H) | 15 min | | | 30 min | | | 60 min | | |
|---|---|---|---|---|---|---|---|---|---|
| | MAE | RMSE | MAPE | MAE | RMSE | MAPE | MAE | RMSE | MAPE |
| Graph-WaveNet (Wu et al., 2019) | **2.46** | **4.42** | **5.44%** | 3.14 | 5.89 | 7.75% | 3.81 | 7.23 | 10.50% |
| GMAN (Zheng et al., 2020) | 2.72 | 5.50 | 6.28% | 3.34 | 6.76 | 8.42% | 3.88 | 7.76 | 10.37% |
| MegaCRN (Jiang et al., 2023) | 2.47 | 4.44 | 5.62% | 3.19 | 6.17 | 8.34% | 3.82 | 7.48 | 10.76% |
| TESTAM | **2.46** | 4.52 | _5.48%_ | **3.10** | **5.75** | **7.62%** | **3.69** | **7.16** | **9.96%** |

| PEMS-BAY (E) | 15 min | | | 30 min | | | 60 min | | |
|---|---|---|---|---|---|---|---|---|---|
| | MAE | RMSE | MAPE | MAE | RMSE | MAPE | MAE | RMSE | MAPE |
| Graph-WaveNet (Wu et al., 2019) | 2.64 | 4.73 | 6.22% | **3.39** | 6.21 | 8.91% | 4.20 | 7.78 | 12.69% |
| GMAN (Zheng et al., 2020) | 2.82 | 5.11 | 6.99% | 3.55 | 6.68 | 9.72% | 4.19 | 7.83 | 12.23% |
| MegaCRN (Jiang et al., 2023) | 2.61 | 4.65 | 6.25% | 3.51 | 6.70 | 10.01% | 4.24 | 8.14 | 13.11% |
| TESTAM | **2.59** | **4.58** | **5.98%** | **3.39** | **6.15** | **8.82%** | **4.03** | **7.57** | **11.45%** |

| EXPY-TKY (I) | 10 min | | | 30 min | | | 60 min | | |
|---|---|---|---|---|---|---|---|---|---|
| | MAE | RMSE | MAPE | MAE | RMSE | MAPE | MAE | RMSE | MAPE |
| Graph-WaveNet (Wu et al., 2019) | 8.28 | 12.56 | 28.65% | 9.40 | 14.23 | 33.69% | 10.24 | 15.35 | 36.99% |
| GMAN (Zheng et al., 2020) | 8.14 | 12.45 | 28.81% | 8.75 | 13.43 | 31.11% | 9.26 | 14.20 | 32.62% |
| MegaCRN (Jiang et al., 2023) | 8.06 | 12.30 | 27.94% | 8.98 | 13.71 | 32.22% | 9.64 | 14.63 | 35.07% |
| TESTAM | **7.87** | **12.26** | **26.95%** | **8.60** | 13.44 | **29.69%** | **9.03** | **14.06** | **31.83%** |

| EXPY-TKY (H) | 10 min | | | 30 min | | | 60 min | | |
|---|---|---|---|---|---|---|---|---|---|
| | MAE | RMSE | MAPE | MAE | RMSE | MAPE | MAE | RMSE | MAPE |
| Graph-WaveNet (Wu et al., 2019) | 8.45 | 12.79 | 30.97% | 9.63 | 14.50 | 36.07% | 10.52 | 15.64 | 39.75% |
| GMAN (Zheng et al., 2020) | 8.30 | 12.63 | 31.08% | 8.90 | 13.57 | 33.58% | 9.44 | 14.33 | 35.19% |
| MegaCRN (Jiang et al., 2023) | 8.24 | 12.49 | 30.47% | 9.21 | 13.92 | 34.60% | 9.90 | 14.87 | 38.00% |
| TESTAM | **8.06** | **12.48** | **29.08%** | **8.81** | 13.67 | **31.88%** | **9.26** | **14.31** | **34.23%** |

| EXPY-TKY (E) | 10 min | | | 30 min | | | 60 min | | |
|---|---|---|---|---|---|---|---|---|---|
| | MAE | RMSE | MAPE | MAE | RMSE | MAPE | MAE | RMSE | MAPE |
| Graph-WaveNet (Wu et al., 2019) | 11.18 | 16.23 | 65.97% | 11.93 | 17.21 | 71.71% | 12.32 | 17.69 | 74.75% |
| GMAN (Zheng et al., 2020) | 11.02 | 16.06 | 64.93% | 11.30 | 16.47 | 66.84% | **11.49** | 16.74 | 67.10% |
| MegaCRN (Jiang et al., 2023) | 11.04 | 16.32 | **62.13%** | 11.67 | 17.07 | 67.93% | 11.94 | 17.37 | 71.43% |
| TESTAM | **10.82** | **15.94** | _63.30%_ | **11.28** | **16.01** | **65.51%** | **11.49** | **16.46** | **66.94%** |

## C.1 Qualitative Evaluation

We perform a qualitative evaluation on TESTAM by visualizing the impact of our context-aware spatial modeling in four different types of cases: 1) hard-to-predict roads with recurring patterns; 2) isolated roads (I); 3) roads with unique traffic patterns; and 4) roads with non-recurring patterns for evaluating the event awareness of TESTAM. We use the EXPY-TKY dataset, which contains complex urban road networks with various traffic patterns.

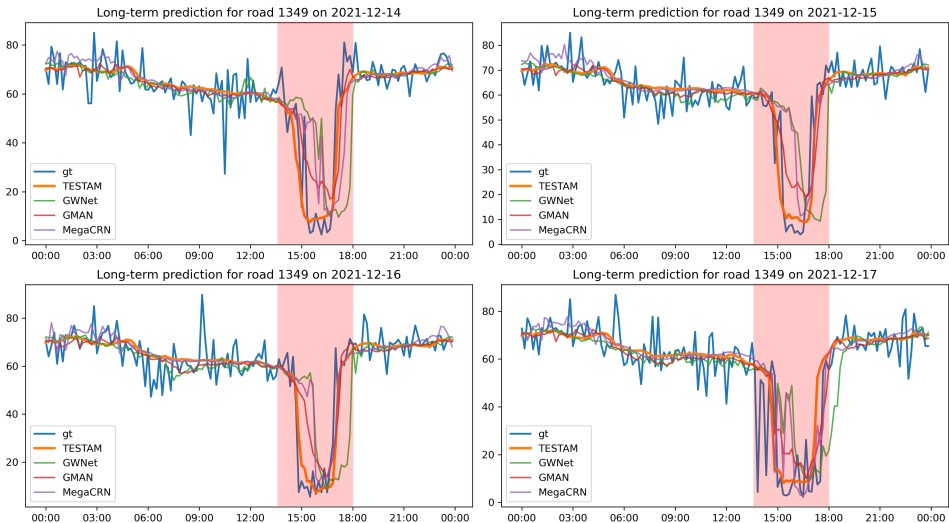

Figure 2: Visualization of a recurring pattern in a hard-to-predict road, Road 1349, from Dec 14th to Dec 17th. Road 1349 is a highway entrance located near Tokyo station. The locations of the roads are indicated in Fig. 7.

**Recurring Patterns on Hard-to-Predict Roads**   With the hard-to-predict roads, we observe that previous models often fail to effectively encode the spatial and temporal correlation of the roads, as Fig. 2 shows. For example, Road 1349 is a highway entrance located near the Tokyo station that has one of the largest traffic volumes in Tokyo and accordingly has severe fluctuation in data. Because of the fluctuations and complex spatio-temporal dependencies of the roads, prior models have shown their limitations in spatial modeling. In particular, Graph-WaveNet (the green line in Fig. 2), which relies on the learnable static graph, fails to timely catch both the speed drop and rise in the red box of Fig. 2. However, GMAN and MegaCRN (the red and violet lines) properly model the ends of rush hour, but they also fail to predict the start of rush hour. Furthermore, MegaCRN exhibits noise-sensitive behavior on Dec. 14th (top-left) and 17th (bottom-right) in Fig. 2.

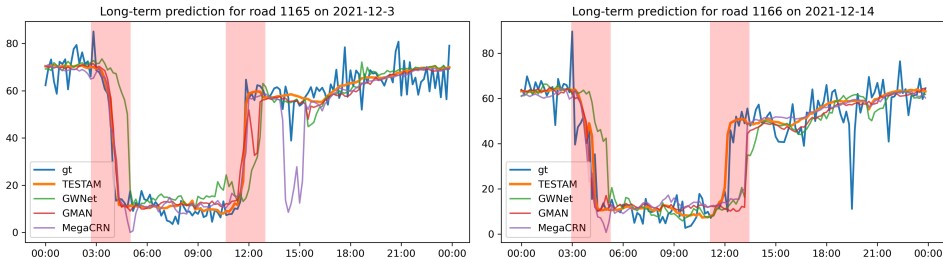

Figure 3: Qualitative forecasting result analysis for spatially isolated roads (I). The locations of the roads are indicated in Fig. 6.

**Spatially Isolated Roads (I)**   When forecasting spatially isolated roads, the model should focus on the road itself, instead of referring to the other roads, which is less informative for prediction.

However, since existing models have less consideration for the importance of self-referencing, they fail to properly model rapid speed changes (e.g., noon in Fig. 3), or they become confused by information the other roads (MegaCRN at 15:00 on Dec 3rd in Fig. 3), resulting in poor forecasting. In contrast, TESTAM accurately forecasts the rapid speed changes that occurred at 3:00 and 12:00, as it enhances temporal modeling with identity experts, temporal information embedding, and a time-enhanced attention layer.

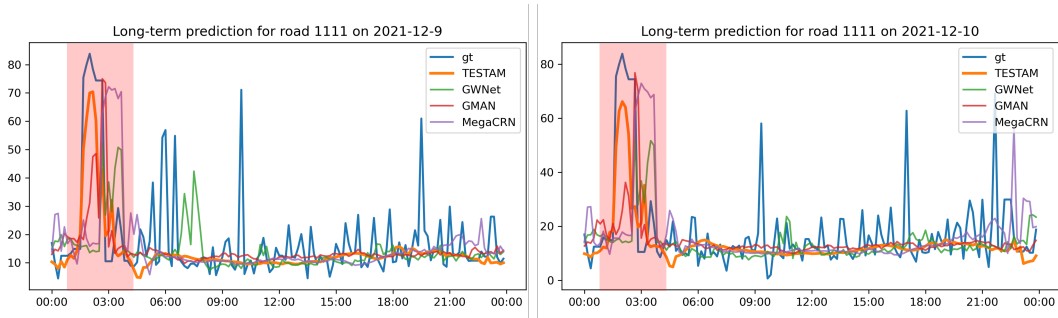

Figure 4: Qualitative forecasting result analysis for Road 1111, a highway ramp located in Shibuya, with unique traffic patterns. The locations of the roads are indicated in Fig. 7.

**Roads with Unique Traffic Patterns: The case of a Highway Ramp**   In the EXPY-TKY, there are many roads showing unique patterns due to complex urban road networks and various traffic behaviors (e.g., commuting and traveling). Road 1111 is a highway ramp located in Shibuya and has unique patterns. One unique pattern of the road is that it tends to have lower speed than 30km/h for all day, except 3:00 (the red box in Fig. 4). GMAN and Graph-WaveNet cannot handle such unique patterns properly, failing to model the increasing speed at 3:00. On the other hand, MegaCRN predicts a high-speed situation by its pattern-awareness of memory units, but it still fails to timely forecast it. In contract, TESTAM timely forecasts the traffic changes, revealing its superiority for modeling the unique behaviors of roads, as shown in Fig. 4.

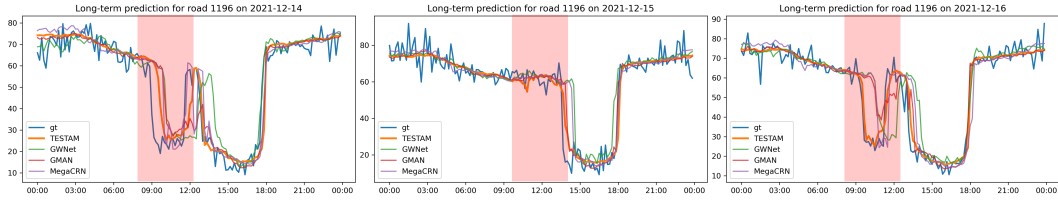

Figure 5: Qualitative analysis results for Road 1196 (a metropolitan expressway) on Dec. 14th, where traffic control may occur because of heavy snow (red boxes)

**Event-Aware Forecasting Case**   We qualitatively evaluate and show the importance of context-aware spatial modeling in improving forecasting performance in various traffic conditions, such as sudden traffic control. Fig. 5 visualizes the recurring and non-recurring traffic conditions caused by traffic control for heavy snow. Because of unexpected traffic controls, there exist sudden speed drops from morning to noon. In such a non-recurring traffic condition, TESTAM shows better forecasting results due to its context-awareness. GMAN and MegaCRN partially capture the sudden changes but cannot make timely predictions.

# D   DETAILED SELECTION PROCEDURES AND LOCATIONS OF THE ROADS FOR CASE STUDY

In this section, we describe how we selected roads and time for each scenario, (I), (H), and (E). In the cases of (I) and (H), we extracted the roads regardless of time.

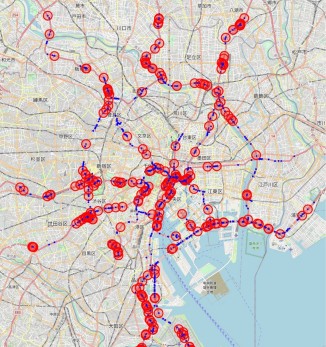 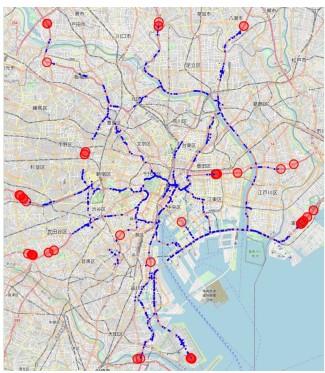 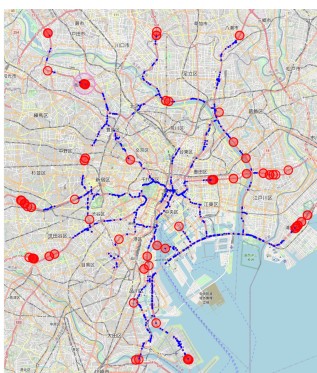

Figure 6: Location visualization for spatially isolated roads. The read circles are selected roads and the blue circles are unselected ones. (Left) the 262 roads before filtering, (middle) newly selected roads from visual investigation, and (right) finalized list of 162 roads. The pink circle is location of Road 1165 and 1166.

**Spatially Isolated Roads (I)** We have selected spatially isolated roads with two procedures: 1) investigating network topology; and 2) filtering out them by visually investigating their locations. From the investigation of network topology, we found that there are total 262 roads without any connection. However, the random sampling process for building traffic dataset makes the network topology be sparse, which could not fully represent the connectivity in the real-world. Therefore, instead of directly utilize total 262 roads, we filter them out by visual investigation. After visual investigation, we have filtered out 150 roads from the list and added 50 roads to the list. Finally, we have total 162 roads for (I), as shown in Fig. 6.

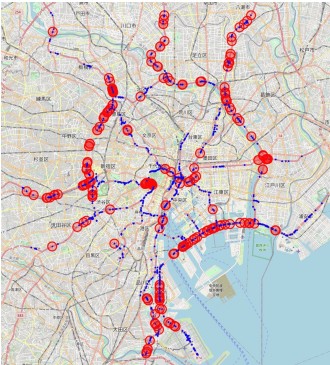 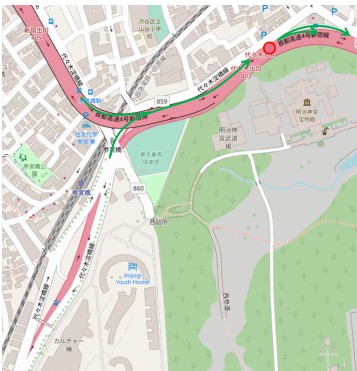 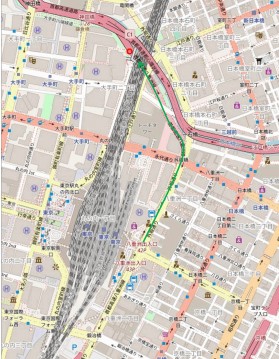

Figure 7: Location visualization for hard-to-predict roads (left) and specific location of Road 1111 (middle) and 1349 (right). The green arrows indicate main traffic flows and directions for each road.

**Hard-to-Predict Roads (H)** For the hard-to-predict roads (H), we conduct two selection processes, visual exploration and entropy-based time-series complexity. Inspired by Ryan et al. (2019), we use entropy-based time-series complexity, which could measure noise level and unpredictability of changes in a series of points by estimating the probability of whether similar patterns will be repeated. From whole set of 1843 roads, we have extracted the 184 roads with the top 10% largest entropy, which are the most unpredictable roads. Furthermore, we additionally insert 34 hard-to-predict roads, such as intersections, ramp, and highway entrances by visually investigating roads. We finalized our list as shown in Fig. 7 left.

**Roads and Time with Sudden Events (E)** The circumstances in (E) are especially important, since they are location- and time-specific events, and finding those samples requires tremendous efforts. In this paper, we have found the events with two strategies: 1) find specific time intervals (e.g., Christmas) of hard-to-predict roads (Fig. 7 left); and 2) find traffic controls and constructions that

are relatively easy to find than accidents and have credible source, including official announcements from the Metropolitan Expressway Co., Ltd[1]. As a result, we have chosen the data at holiday intervals in hard-to-predict roads and the roads and intervals in construction or sudden traffic controls for (E).

---

[1]https://www.shutoko.co.jp/

