# OpenReview forum: "TESTAM: A Time-Enhanced Spatio-Temporal Attention Model with Mixture of Experts"
_ICLR.cc/2024/Conference — ICLR 2024 poster_

### Official Review · Reviewer_SSH9 · 2023-10-19

**Soundness:** 2 fair
**Presentation:** 2 fair
**Contribution:** 3 good
**Rating:** 6
**Confidence:** 4

**Summary:**

The authors point out that accurate traffic forecasting requires capturing two distinct traffic patterns, recurring and non-recurring. To accomplish this goal, they propose TESTAM, a mixture-of-expert model with three heterogeneous experts responsible for different aspects of spatio-temporal modeling. By properly routing the experts, the model learns to handle various circumstances adaptively. Experimental results on three benchmark datasets verify the effectiveness of TESAM.

**Strengths:**

1. The manuscript is well-organized and easy to follow.
2. The introduction of mixture-of-expert models increases the capacity of spatial modeling for traffic forecasting, and to some extent, increases the interpretablility.

**Weaknesses:**

1. In terms of spatial and temporal modeling, this manuscript does not seem to bring new techniques.
2. There are multiple sets of terminologies describing the same thing without an explicit connection. For example, when categorizing traffic patterns, both "recurring vs non-recurring" and "normal vs abnormal" (in Conclusion) are used; when describing the three experts, "temporal modeling, spatio-temporal modeling with static graph, spatio-temporal modeling with dynamic graph" (in Abstract) and "identity experts, adaptive experts, attention experts" (in Figure 1). Clarifying these key concepts will make the presentation clearer.
3. Only the overall metrics for predicting accuracy are reported. It is unclear what recurring and non-recurring patterns that TESTAM captured while other methods miss. Some in-depth experimental analysis (see Q4 for details) will better support the claim of the first contribution.

**Questions:**

1. Why the proposed method is abbreviated as TESTAM?
2. What does "in-situ" mean in the context of this manuscript?
3. The temporal information embedding and time-enhanced attention seem novel for traffic forecasting. Do they affect the performance? Some discussion or ablation study is needed.
4. In the abstract, the authors state that TESTAM can model isolated nodes, highly related nodes, and recurring and non-recurring events. Can you provide some experimental results and analysis for this claim, e.g. case study?

Minor:
1. Please add comments to the meaning of results in bold and underlined in Table 1.
2. Descriptions and analysis of the qualitative examples in the supplementary material are missing.
3. Why are some equations in section 3 not numbered? I believe some of them are proposed by the authors, e.g. time-enhanced attention, and thus should be numbered.
4. Please provide an explicit formula for the best-route loss, at least in the supplementary material.

---

> ### Author Response · Authors · 2023-11-22
> **Official Response to the Reviewer SSH9**
>
> We are grateful to receive such valuable comments. We improved our manuscripts accordingly. We provide detailed responses below.
>
> W1. Lack of novelty in terms of spatial and temporal modeling
> > Thank you for your feedback! We would like to emphasize that this paper proposes novel context-aware spatial modeling (i.e., different spatial modeling method selections), which reveals that spatio-temporal modeling with multiple spatial modeling methods could lead to better forecasting results.
>
> > Furthermore, Time-enhanced attention and temporal information embedding are our approach’s novel temporal modeling methods, which enable better information transfer from the input domain to the output domain. As shown in Table 2 in Sec. 4, without time-enhanced attention, TESTAM experiences a performance degradation from 6.40 to 6.64 in terms of MAE with EXPY-TKY. Please see Table 2 in Sec. 4 and check the ablation study results.
>
> W2. Inconsistent usage of terms
> > Thank you for your feedback! We have revised the inconsistent use of terms and clarified them in our manuscript.
>
> W3, Q4. Only the overall metrics are reported, authors should conduct an in-depth experimental analysis.
> > Thank you for your valuable comment! We additionally provide the newly updated and detailed experimental results in Appendix C. Based on the experiment results for roads that have difficult conditions–isolated roads, intersections, or sudden traffic jams caused by events (e.g., accidents)–TESTAM achieves a consistent performance improvement of 4% to 7% compared to other models. We also provide qualitative results in Appendix C with descriptions that showcase how TESTAM improves performance under difficult conditions.
>
> Q1. Why the proposed method is abbreviated as TESTAM?
> > TESTAM is the abbreviation for “time-enhanced spatio-temporal attention model.” We clarified this in the Introduction.
>
> Q2. What does "in-situ" mean in the context of this manuscript?
> > We use the term “in-situ” to better describe the dynamic and context-aware selection of spatial modeling methods.
>
> Q3. Do temporal information embedding and time-enhanced attention affect the performance?
> > We newly updated our ablation study results (i.e., Table 2 in Sec. 4) for TIM and time-enhanced attention. Without time-enhanced attention, the model indicates high-performance degradation from 6.40 to 6.64 in terms of MAE for the EXPY-TKY dataset.
>
> We also resolved minor issues. Here is the list:
> 1. We clarified the meaning of the results in bold and underlined in Table 1. Results in bold indicate the best performance and those underlined indicate the second-best performance
> 2. We updated the qualitative analysis and descriptions in Appendix, C.
> 3. We revised manuscripts for the equations to be numbered
> 4. We provide explicit formula and descriptions for pseudo labels in Appendix A

---

> ### Comment · Reviewer_SSH9 · 2023-11-22
>
> Your response addresses most of my concerns. Some minor questions are appended as follows.
>
> W2. I have noticed some efforts of the authors to unify the terminologies. But what are the three experts responsible for respectively, temporal modeling, spatio-temporal modeling with static graph, spatio-temporal modeling with dynamic graph? They first appear in Figure 1 without explanation, and their effects are discussed in Section 4.3. Clarification is needed.
>
> W3. The added experimental results substantially increase the quality of this manuscript. It would be better to report some statistics of the three scenarios (I, H, E) to help readers gain an intuitive understanding. In Appendix C.1, the authors can visualize the selected roads if possible.

---

> > ### Author Response · Authors · 2023-11-23
> > **Official Response to Reviewer SSH9**
> >
> > Thank you for the additional comments! We clarified detailed information in our manuscript. Below are our answers.
> >
> > W2. What are the three experts responsible for respectively?
> >
> > > We additionally provide information in Fig. 1, as "Identity, adaptive, and attention experts are responsible for temporal modeling, spatial modeling with a learnable static graph, and with a dynamic graph (i.e., attention), respectively." We would like to kindly inform you that we additionally provide the description of the spatial modeling methods in Sec. 3.2 (in the paragraph  "spatial modeling layer").
> >
> > W3-1. Could the authors report some statistics of three scenarios?
> >
> > > Thank you for your comment! The two scenarios, (I), and (E), have been chosen with visual investigation and based on news articles, respectively, so we do not have statistical information for them. However, we provide a qualitative analysis of them in Appendix C.1, especially in the paragraphs "Spatially Isolated Roads (I)" and "Event-Aware Forecasting Case."
> >
> > > For (H), we additionally mentioned that we selected 1) the intersections or ramps based on our visual investigation and 2) the roads with the top 10\% entropy-based time-series complexity.
> >
> >
> > W3-2. The authors can visualize the selected roads if possible
> >
> > > Thank you for your comment! If we have simultaneously visualized the location of the selected roads and line charts associated with the roads, it may reduce readability with smaller line charts and too much information provided at once.

---

> ### Comment · Reviewer_SSH9 · 2023-11-23
>
> Based on the authors' response, I decide to raise my rating to 6. Still, I have some suggestions for Appendix C.
>
> W3-1. The numbers of selected roads in three scenarios can be reported. The procedure of selecting these roads can be described more precisely. For example,
>
> (I) How to quantitatively select isolated roads based on the adjacency matrix?
>
> (H) What is the top 10% entropy-based time-series complexity?
>
> (E) Please provide references to the news articles that help identify roads with sudden events.
>
> W3-2. The road nets and the line chart can be visualized separately. The authors may visualize some selected roads from the three scenarios.

---

> > ### Author Response · Authors · 2023-11-23
> > **Official Response to Reviewer SSH9**
> >
> > Thank you for your detailed feedback and sorry for the last-minute response. Based on your feedback, we have noticed that we could improve our manuscripts by providing detailed information on how to choose the roads for scenarios. We additionally describe those in Appendix D. Below are the improvements we made.
> >
> > W3-1. Precise description of the selection process
> > > We provide the selection processes for each scenario (I), (H), and (E) in Appendix D, including the number of selected roads. For (I), we have chosen 162 roads, which are about 10\% of the whole roads. In the case of (H), we have selected 184 roads using time-series complexity and include 34 hard-to-predict roads, such as intersections, ramps, and highway entrances. For (E), we have selected two circumstances: 1) Christmas and Eve in the hard-to-predict roads (H) and 2) traffic control and construction reported from the official announcements.
> >
> > W3-2. The road net can be visualized separately.
> > > Thank you for your suggestion! In addition to the selection process, we have updated the locations in Fig. 6 and 7. We especially provide the locations of the roads, including Road 1111, 1165, 1166, and 1349. Please note that we did not report the location of 1349 since it is the normal metropolitan expressway with a sudden event (i.e., traffic control).

---

### Official Review · Reviewer_FrvU · 2023-10-25

**Soundness:** 3 good
**Presentation:** 2 fair
**Contribution:** 2 fair
**Rating:** 5
**Confidence:** 4

**Summary:**

In this paper, authors introduce TESTAM, a novel Time-Enhanced Spatial-Temporal Attention Model to realize accurate traffic forecasting. TESTAM is a Mixture-of-Experts model that incorporates attention mechanisms, allowing real-time spatial modeling under both normal and abnormal circumstances. By reformulating the routing problem as a classification task, TESTAM can adapt to different traffic conditions, enabling the selection of appropriate spatial modeling methods.

**Strengths:**

- S1. Considering the complex dependency on road networks, the use of an expert mixture model in spatial-temporal forecasting is an innovative design.

- S2. The paper provides a clear categorization of spatial modeling types for spatial-temporal forecasting, and the selection of different spatial models (identity experts, adaptive experts, attention experts) is highly representative.

**Weaknesses:**

W1. There are many typos, and some incorrect writings can seriously mislead the readers:
   - '...choose spatial modeling methods (i.e., expert) properly; and' on page one.
   - $f(X^{(t)},\theta)$ should be replaced by 'g(X^{(t)},\theta)' on page three.
   - In the formulas at the bottom of page five, the superscript (k) in $W_{q}^{(k)}$ and $W_{k}^{(k)}$ appears to be meaningless and should be replaced by $W_q$ and $W_k$.
   - On the fifth page, $\[\tau^{(t+1),...,\tau^{(t+T)}\]$ should be changed to $\[\tau^{(t+1),...,\tau^{(t+T')}\]$.
   - In formula 5 on the sixth page, 'larger than q-th quantile' should be corrected to 'smaller than q-th quantile.'
   - In the seventh page, the sentence 'excludes the relatively coarse-grained loss function only' should have 'only' removed.
   - The functions such as 'Concat' and the activation functions 'relu' should be written as upright letters.
   - Some formulas have equation numbers, while others do not.

W2. The effectiveness of Time-enhanced Attention has not been verified through ablation experiments, and readers cannot determine whether it truly enhances the model's predictive performance.

W3. The distinction between best-route selection and worst routing avoidance is not clear. When introducing best-route selection, the authors mention node-wise routing and node-wise pseudo-label but haven't explained them. Additionally, changing 'smaller than q-th quantile' to 'smaller than (1-q)-th quantile' alone is considered best-route selection, but the reason for this change has not been provided.

W4. Experimental results:
   - The experimental results are inconsistent. In Table 2, the results for TESTAM in the ablation experiment are for a 30-minute forecast, but they do not match the results in Table 1. Additionally, I noticed that in Table 2, the PEMS-BAY dataset performs better in the 'w/o gating' condition than the complete TESTAM model in Table 1.

   - The magnitude of the performance improvement is minimal. Table 1 shows that many metrics only slightly improve in terms of percentiles, at the cost of three times the computational time and memory for building three expert models.

W5. The title doesn’t well match with this study itself. Actually, this work mostly focuses on traffic forecasting, but the title   suggesting a new general Spatio-Temporal Attention Model with MoE seems much more general and broader.

**Questions:**

Q1- Q2: Please refer  to W3 and W4.
Q3. In memory-based gating networks, why the similarity between the model's input and output is used as the routing probability? This lacks a certain theoretical foundation (or some analysis) and explanation.

**Details Of Ethics Concerns:**

NA.

---

> ### Author Response · Authors · 2023-11-22
> **Official Response to the Reviewer FrvU**
>
> Thank you for your detailed comments! Those reviews are really helpful to improve our manuscripts. Below are detailed responses to questions and how we improved our manuscripts.
>
> W1. Many typos, incorrect writing, and grammar errors
> > We first revised the manuscript with authors to fix the typos related to mathematical equations and worked with native English speakers to improve our manuscripts.
>
> W2. The effectiveness of time-enhanced attention should be verified through ablation study
> > We newly updated our ablation study results (i.e., Table 2 in Sec. 4) for time-enhanced attention. Without time-enhanced attention, the model indicates high-performance degradation from 6.40 to 6.64 in terms of MAE for the EXPY-TKY dataset.
>
> W3. Need detailed information for the best-route selection loss function
> > We updated the detailed information on the best-route selection in Appendix A. In our paper, we set q = 0.7. In this setting, an error greater than 70% of the error distribution, which is equal to the top 30% MAE, will be the worst route. In this setting (q=0.7), the bottom 30% MAE will be the best route, which means an error smaller than 1 - q-th quantile will be the best route. We revised our manuscripts and added detailed explanations in Appendix A.
>
> W4-1. About experimental results in Table 2
> > Our results are actually consistent. To further clarify this, the results in Table 2 are reported across all forecasting windows. This means that the averages from short-term to long-term forecasting results. We have clarified the missing information in our manuscript.
>
> W4-2. Concerning computational costs
> > TESTAM uses a small number of parameters with fewer layers compared to other models, as a result, the computational costs are not expensive. We report a detailed time complexity analysis in Appendix B. For your information, we briefly summarize the results here. TESTAM requires a similar time as DCRNN in training. In the inference phase, TESTAM shows the second fastest computation (about 7 seconds) with the smallest number of the parameters, which is faster than that of the other models except Graph-WaveNet (about 4 seconds).
>
>
> W4-3. Marginal improvements in Table 1, require a case study and investigation
> > As we described in our manuscripts, we focus on in-situ selection in spatial modeling methods to improve situation awareness of the model. We additionally provide the newly updated and detailed experimental results in Appendix C. Based on the experiment results for roads that have difficult conditions–isolated roads, intersections, or sudden traffic jams caused by events (e.g., accidents)–TESTAM achieves a consistent performance improvement of 4% to 7% compared to other models. We also provide qualitative results in Appendix C with descriptions that showcase how TESTAM improves performance under difficult conditions.
>
> W5. The title isn't well-matched to the manuscripts
> > Thank you! We will add the traffic forecasting in the title in the final version. It will be "TESTAM: A Time-Enhanced Spatio-Temporal Attention Model with Mixture of Experts for Traffic Forecasting"
>
> Q3. Why do the authors utilize the similarity between input and output as routing probability?
> > Thank you for your feedback. I guess there exists a misunderstanding. In short, the similarity between $z_e$ and $O_i$ is equivalent to that between the output representation of experts and input-conditioned memory units trained to be in the same space as $z_e$.
>
> > In our design, we first make query $Q_i$ by projecting input to the query domain. In this process, the projection learns the input-output mapping, which converts input $X$ (in the input domain) into the $Q$ (in the same domain as final hidden states). Afterward, it queries the memory unit by attention mechanism. Then, the queried memory $O$ is an input-conditioned memory unit that will be trained to be in the same space as the output representations of experts. Therefore, in fact, the similarity between $z_e$ and $O_i$ is equivalent to one between the output representation of experts and the input-conditioned output representation. In this design, the routing probability will be interpreted as the finding of the most feasible solution from experts by using input-conditioned typical output representation.

---

> > ### Comment · Reviewer_FrvU · 2023-11-22
> > **Thanks for author rebuttal**
> >
> > Thanks for your time and efforts on answering above questions.  After reading other reviews and author rebuttal, I lean to remain my score.

---

### Official Review · Reviewer_gqdu · 2023-10-31

**Soundness:** 3 good
**Presentation:** 3 good
**Contribution:** 2 fair
**Rating:** 6
**Confidence:** 4

**Summary:**

This paper presents a mixture of experts model for traffic forecasting where each expert model uses a different spatial correlation learning model. In addition, a time-enhanced attention is added to help model the temporal correlation on top of the usual attention module. Experiments haven been conducted on three datasets, two of which of commonly used benchmark datasets. There are 13 baseline models, including one from 2022 and one from 2023.

**Strengths:**

1. While the core of the proposed model is based on mixture of experts, there are new designs to the components including an extra time-enhanced attention module and an enhanced routing classification loss considering the best-routing selection.

2. The paper is very well written overall and is easy to follow.

3. There are quite a few baseline models including some of the latest.

**Weaknesses:**

The main issue of the paper is perhaps the relatively weak experimental results. As shown in Table 1, the proposed model TESTAM has very similar performance to MegaCRN (Jiang et al., 2023) and sometimes PM-MemNet (Lee et al., 2022).

There are no model training time results. I wonder if TESTAM is even slower than MegaCRN and/or PM-MemNet to train.

The discussion of the results have focused on comparisons with GMAN and StemGNN which are from 2020 and are relatively "old" methods already.

Overall, the importance of the proposed techniques has not been shown with strong evidence.

Minor presentation issues:
Typo: "is noisy sensitive", "ahve", "Let define";
Check "where f(X(t), θ)" below Equation 1;
Grammar: "traffic data is noisy and contains many non-stationary, make the best route selection hard."

**Questions:**

See the weak points.

---

> ### Author Response · Authors · 2023-11-22
> **Official Response to the Reviewer gqdu**
>
> Thank you for your detailed and valuable feedback! We revised and added additional analysis based on your comments. Below are the detailed responses to your comments.
>
>
> W1, W3, and W4, Weak experimental results, less discussion on state-of-the-art models, and need for evidence to support the importance of proposed methods
> > As we described in our manuscripts, we focus on in-situ selection in spatial modeling methods to improve situation awareness of the model. We additionally provide the newly updated and detailed experimental results in Appendix C. Based on the experiment results for roads that have difficult conditions–isolated roads, intersections, or certain events, such as sudden traffic jams by accidents–TESTAM achieves a consistent performance improvement of 4% to 7% compared to other models, including MegaCRN, which is a state-of-the-art model. We also provide qualitative results in Appendix C with descriptions that showcase how TESTAM improves performance under difficult conditions.
>
> W2. Queries for the computational cost analysis
> > TESTAM uses a small number of parameters with fewer layers compared to other models, as a result, the computational costs are not expensive. We report a detailed time complexity analysis with the METR-LA dataset in Appendix B. For your information, we briefly summarize the results here. TESTAM requires a similar time as DCRNN in training. In the inference phase, TESTAM shows the second fastest computation (about 7 seconds) with the smallest number of the parameters, which is faster than that of the other models except Graph-WaveNet (about 4 seconds).
>
> W5. Typos and grammar errors
> > We first revised the manuscript with authors to fix the typos related to mathematical equations and worked with native English speakers to improve our manuscripts.

---

### Official Review · Reviewer_orKb · 2023-11-01

**Soundness:** 3 good
**Presentation:** 3 good
**Contribution:** 3 good
**Rating:** 6
**Confidence:** 3

**Summary:**

This paper proposes a novel deep learning model named TESTAM, which individually models recurring and non-recurring traffic patterns by a mixture-of-experts model with three experts on temporal modeling, spatiotemporal modeling with static graph, and dynamic spatio-temporal dependency modeling with dynamic graph.
By introducing different experts and properly routing them, TESTAM could better model various circumstances, including spatially isolated nodes, highly related nodes, and recurring and non-recurring events.
The evaluation is conducted with three open datasets and compares the proposed method and existing methods.

**Strengths:**

The idea of the proposed method, which utilizes multiple experts models and switches between them adaptively to deal with various traffic conditions, is reasonable and interesting for forecasting urban traffic affected by various factors.

The proposed method's level of prediction compared to existing techniques is demonstrated by utilizing three datasets and comparing it to a number of existing methods.

The structure of the paper is easy to understand and the individual contents are clearly described.

**Weaknesses:**

Table 1 shows that the error is small in many conditions compared to existing methods, but the difference is not large, and it is not clear what the significance of this error is.

It is also not clear where the computational cost of the proposed method in model building and forecasting stands in comparison to other methods. In my opinion, this is important information, especially how the proposed method relates in terms of computational cost to methods that do not have very large differences in accuracy.

It is not clear how the proposed method is able to deal with the "various circumstances" described in the intro of the paper.

**Questions:**

Once the information pointed out in Weakness is clarified, the validity of the proposed method will become clearer.
Could the authoer please clarify these points?

---

> ### Author Response · Authors · 2023-11-22
> **Official Response to the Reviewer orKb**
>
> Thank you for your detailed review! We revised our paper accordingly. Below are the detailed responses and revisions.
>
> W1 and W3. Table 1 shows marginal improvements and only reports the averaged performance without reporting the performances in various circumstances, including recurring and non-recurring traffic patterns.
> > Thank you for your feedback! As we described in our manuscripts, we focus on an in-situ selection of spatial modeling to improve situation awareness of the model. To emphasize the contribution and validate the situation-awareness of TESTAM, we additionally provide the detailed experimental results in Appendix C. Specifically, in the case of isolated roads, intersections, or sudden traffic jams caused by events (e.g., accidents), TESTAM achieves a consistent improvement of 4% to 7% compared to existing models.
>
> W2. Queries for the computational costs analysis
> > TESTAM uses a small number of parameters with fewer layers compared to other models, as a result, the computational costs are not expensive. We report a detailed time complexity analysis with the METR-LA dataset in Appendix B. For your information, we briefly summarize the results here. TESTAM requires a similar time as DCRNN in training. In the inference phase, TESTAM shows the second fastest computation (about 7 seconds) with the smallest number of the parameters, which is faster than that of the other models except Graph-WaveNet (about 4 seconds).

---

### Author Response · Authors · 2023-11-22
**Response to Common Questions**

We thank the reviewers for their valuable comments. We believe that we have further improved our manuscripts accordingly.

In these comments, we provide responses to the common questions raised by reviewers.

### **The improvements in Table 1 seem marginal – need more detailed information to support TESTAM is really effective for various circumstances**
> As we described in our manuscripts, we focus on in-situ selection in spatial modeling methods to improve situation awareness of the model. We additionally provide the newly updated and detailed experimental results in Appendix C. Based on the experiment results for roads that have difficult conditions–isolated roads, intersections, or sudden traffic jams caused by events (e.g., accidents)–TESTAM achieves a consistent performance improvement of 4% to 7% compared to other models. We also provide qualitative results in Appendix C with descriptions that showcase how TESTAM improves performance under difficult conditions.

### **Provide detailed information for best-route selection loss and time complexity of TESTAM**
> (Computational costs) TESTAM uses a small number of parameters with fewer layers compared to other models, as a result, the computational costs are not expensive. We report a detailed time complexity analysis in Appendix B. For your information, we briefly summarize the results here. TESTAM requires a similar time as DCRNN in training. In the inference phase, TESTAM shows the second fastest computation (about 7 seconds) with the smallest number of the parameters, which is faster than that of the other models except Graph-WaveNet (about 4 seconds).

> (About best-route selection) We updated the detailed information on the best-route selection in Appendix A. In our paper, we set q = 0.7. In this setting, an error greater than 70% of the error distribution, which is equal to the top 30% MAE, will be the worst route. In this setting (q=0.7), the bottom 30% MAE will be the best route, which means an error smaller than 1 - q-th quantile will be the best route. We revised our manuscripts and added detailed explanations in Appendix A.


### **Effectiveness of TIM and time-enhanced attention**
> We newly updated our ablation study results (i.e., Table 2 in Sec. 4) for TIM and time-enhanced attention. Without time-enhanced attention, the model indicates high-performance degradation from 6.40 to 6.64 in terms of MAE for the EXPY-TKY dataset.


### **Fix typo and grammar errors**
> We worked with native English speakers to improve our manuscripts.

---

### Meta-Review · Area_Chair_PHJT · 2023-12-08

**Metareview:**

The paper presents TESTAM, a Time-Enhanced Spatial-Temporal Attention Model crafted for precise traffic forecasting. Utilizing a Mixture-of-Experts model with three diverse experts addressing distinct aspects of spatio-temporal modeling, TESTAM adapts to varied traffic conditions by reformulating the routing problem and incorporating attention mechanisms. The evaluation, performed on three benchmark datasets, compares TESTAM with existing models, showcasing its effectiveness in capturing recurring and non-recurring traffic.

**Justification For Why Not Higher Score:**

After the rebuttal, the perceived technical novelty or empirical gains are still viewed as somewhat marginal.

**Justification For Why Not Lower Score:**

The reviewers emphasized positive aspects, including the reasonable and interesting idea of the proposed method, its demonstrated prediction performance compared to existing techniques using three datasets, clear and well-organized writing, novel designs in the proposed model, and the innovative use of an expert mixture model in spatial-temporal forecasting. These factors collectively contribute to the recommendation for acceptance (poster).

---

### Decision · Program_Chairs · 2024-01-16

Accept (poster)